# Synbiotic Agents and Their Active Components for Sustainable Aquaculture: Concepts, Action Mechanisms, and Applications

**DOI:** 10.3390/biology12121498

**Published:** 2023-12-06

**Authors:** Vijayaram Srirengaraj, Hary L. Razafindralambo, Holy N. Rabetafika, Huu-Thanh Nguyen, Yun-Zhang Sun

**Affiliations:** 1Fisheries College, Jimei University, Xiamen 361021, China; vijayarambiotech@gmail.com; 2ProBioLab, 5004 Namur, Belgium; probiolab@europe.com; 3BioEcoAgro Joint Research Unit, TERRA Teaching and Research Centre, Sustainable Management of Bio-Agressors & Microbial Technologies, Gembloux Agro-Bio Tech—Université de Liège, 5030 Gembloux, Belgium; 4Department of Biotechnology, An Giang University, Long Xuyen City 90000, Vietnam; nhthanh@agu.edu.vn

**Keywords:** probiotics, prebiotics, synbiotics, gut microbiota, fishes, aquaculture

## Abstract

**Simple Summary:**

Aquatic animals are consistently exposed to the threats of environmental deterioration and infection outbreaks because of the excessive use of antibiotics and synthetic drugs. This practice leads to the accumulation of residues in aquatic systems and the development of antimicrobial resistance among pathogens. Nature-based solutions, such as functional feeds containing synbiotics and their active components, such as probiotics, prebiotics, and postbiotics, play a crucial role in maintaining a healthy environment and promoting the well-being of animals in aquaculture. Drawing upon a thorough literature survey and experimental evidence, these agents have been shown beneficial to aquatic animals and their ecosystems. Consequently, these synbiotic agents and related components emerge as promising natural alternatives to traditional synthetic drugs and antibiotics in aquaculture.

**Abstract:**

Aquaculture is a fast-emerging food-producing sector in which fishery production plays an imperative socio-economic role, providing ample resources and tremendous potential worldwide. However, aquatic animals are exposed to the deterioration of the ecological environment and infection outbreaks, which represent significant issues nowadays. One of the reasons for these threats is the excessive use of antibiotics and synthetic drugs that have harmful impacts on the aquatic atmosphere. It is not surprising that functional and nature-based feed ingredients such as probiotics, prebiotics, postbiotics, and synbiotics have been developed as natural alternatives to sustain a healthy microbial environment in aquaculture. These functional feed additives possess several beneficial characteristics, including gut microbiota modulation, immune response reinforcement, resistance to pathogenic organisms, improved growth performance, and enhanced feed utilization in aquatic animals. Nevertheless, their mechanisms in modulating the immune system and gut microbiota in aquatic animals are largely unclear. This review discusses basic and current research advancements to fill research gaps and promote effective and healthy aquaculture production.

## 1. Introduction

Aquaculture is an emerging sector that generates numerous employment opportunities and also addresses a fundamental need for essential nutrients in global food production [1]. However, it is presently faced with pressing challenges, especially the vulnerability of aquatic animals to ecological degradation and infectious outbreaks. A key contributing factor to these threats is the excessive use of antibiotics and synthetic drugs, which exert harmful effects on the aquatic environment [2,3].

The aquaculture production sector typically relies on traditional practices employing various antibiotics (e.g., chloramphenicol, fluoroquinolones, nitrofurans, quinolones, florfenicol, sufamerazine, chorionic gonadotropin, oxytetracycline dihydrate, and oxytetracycline hydrochloride) and synthetic chemicals (e.g., formalin, malachite green, potassium permanganate, and copper sulfate) to control related diseases [4]. However, some of these chemotherapeutic applications have been widely criticized given their negative impacts on marine debris gathering, drug resistance expansion, and immunosuppressant activity. For example, the use of formalin and potassium permanganate for pathogen control has resulted in adverse effects on fish like damage to gills (hyperplasia) and alteration in mucous cells [5,6]. The extensive application of antibiotics in aquaculture has led to their bioaccumulation in aquatic animals [7]. The intensive use of antibiotics and chemicals leads to the buildup of harmful residues, not only in aquatic animals but also in consumers, by causing side effects such as diarrhea, vomiting, and stomach problems. Moreover, the practice of these traditional methods has been reported to be ineffective in controlling diseases in large-scale aquaculture processes [8,9,10,11,12,13].

In fish, the gastrointestinal tract (GIT) microbiota plays several vital functions. These microbial consortia increase digestive action, enhance the immune system, protect against harmful microbes, and improve intestine development [14]. In recent years, some gnotobiotic (germ-free) animal models have been successfully used as wonderful tools for studying host–microbe interactions and investigating the role of gut microbiota in xenobiotic metabolism [15,16]. Through zebrafish (*Danio rerio*) models, researchers have observed that the presence of alkaline phosphatase in the brush border intestine plays a vital function in gut epithelium division, as well as in the modulation of gene expression in bacteria, which possesses various functional properties (e.g., epithelial maturation, hormone-secreting endocrine organs, and mucous secreting goblet cells) in the gastrointestinal tract in *D. rerio* larvae [17,18]. Recently, it was reported that TLR2/MyD88 signaling plays an essential role in innate immune recognition and activation during the colonization of two indigenous bacteria (*Chryseobacterium* ZOR0023 and *Exiguobacterium* ZWU0009) in zebrafish [19]. Indigenous probiotic strains have significant functions such as developing the immune system (nonspecific and specific immunity) and inducing different types of cytokines, namely, TNF-a, interleukins (IL-6, IL-10, IL-12), and IFN-c [20]. The indigenous probiotic *Bacillus pumilus* SE5 activates the expression of TLR2 signaling and antibacterial peptide genes in the intestine of groupers (*Epinephelus coioides*). Enhanced TLR2 signaling may result from the interaction of the host with the probiotic cell components [21,22]. In order to enhance the immune system in fish, the gut microbiota also provides important protection against pathogenic organisms [23,24].

Functional feed additives such as probiotics, prebiotics, and/or synbiotics in diets have been extensively recommended to maintain a healthy GIT microbial community, improve immunity, and consequently promote the health of cultured aquatic organisms [25,26,27]. These synbiotic- and component-based ingredients, consisting of live microorganisms, inert substrates, and a combination of both, possess a wide range of multiple functionalities. They represent alternative nature-based solutions for improving aquatic animal health and production [24,28,29]. This review provides insights into the basic and current developments in the utilization of probiotics, prebiotics, postbiotics, and synbiotics in aquaculture applications. It also presents a new way to develop a healthy and modern aquaculture industry. 

## 2. Probiotics

### 2.1. Definition and Characteristic Features

The Food and Agriculture Organization (FAO) of the United Nations and the World Health Organization (WHO) define probiotics as “Live microorganisms that, when administered in adequate amounts, confer a health benefit on the host” [30]. Recently, the term probiotics has been associated with microbial feed additives that, when controlled in enough amounts, confer health and beneficial impacts on a host of aquatic animals [29].

Probiotics act as a defense system for the host against harmful microbes or foreign substances [31,32,33,34]. They also produce beneficial bioactive molecules such as enzymes, proteins, lipids, organic acids, and others. Some of these bioactive molecules improve binding to probiotics and reduce, therefore, the activity of pathogens in the gut region through the surface competition mechanism [35]. Probiotics play a significant role in strengthening the immune system of the host [36]. While earlier studies have noted the utilization of probiotics in pigs, poultry, cattle, and humans, their application in aquaculture is a relatively new idea [37,38]. Probiotics can be administered in two ways in aquaculture. They can be supplemented with feed to modulate gut microbes, or they can be directly added to the water, thereby inhibiting the growth of pathogens. These modes of administration are very critical in the utilization of probiotics in aquaculture [39,40]. Probiotics can have alive, dead, or microbial cell components and provide benefits to the host when added to feed or rearing water. This is achieved at least in part by improving the microbial balance of the host or ambient environment [40]. Figure 1 summarizes the different entryways of probiotics and their benefits in the aquaculture system.

Probiotics appear to be a new agent for the development of aquaculture systems, exerting several favorable effects on growth activity, immune systems, digestion, water quality, the inhibition of pathogens, and the regulation of the gut microbes of aquatic animals. The utilization of probiotics in aquaculture is a modern trend, although its effectiveness in the aquatic ecosystem has not been considered comprehensively. Probiotics are ubiquitous, commonly present in aquatic animals, and play an important protective role throughout the digestive system [41,42]. Mainly represented by Lactobacilli, these beneficial microorganisms are vital to preventing illnesses and improving aquatic animal GIT functions by excreting secondary metabolites such as lactic acid and other bioactive compounds [43,44]. These biomolecules, synthesized by probiotics, protect against inhibitory molecules from pathogens [45]. They can also be extracted from probiotics in terrestrial plants and marine life forms and then utilized to enhance disease resistance, develop the immune system, reduce environmental stress, and increase feed quality levels [46,47]. Advanced studies in this field have reported microbial by-product biomolecules such as enzymes, lipids, proteins, and immune toxins [48]. Nowadays, some probiotic products are commercially available and are already used in aquaculture as feed additives [49]. These microbial by-products are beneficial and are mainly helpful in enhancing the health status of aquatic animals.

Potential probiotic strains are assessed based on physiological, functional, and safety criteria such as stress resistance (e.g., acid and bile tolerance), gut epithelial adherence, survival rates, pathogen-inhibiting activities, large-scale cultivability, non-hemolytic activity, non-pathogenicity, the absence of plasmid-encoded antibiotic resistance genes, and beneficial effects on host animals. These include, for instance, their capacity as growth promoters and anti-inflammatory, antimutagenic, and immunostimulatory agents. Each new strain used for probiotic expansion mainly contains all the aforesaid features [28,50,51]. Current and potential probiotic species for use in aquaculture are listed in Table 1.

### 2.2. Possible Modes of Action of Probiotics in Aquaculture

The significant effects of probiotics, e.g., *Bacillus* spp. as feed supplements, include the improvement of growth performance, digestive enzyme activity, resistance to pathogens, and immune responses in aquatic animals [68,69]. Possible action modes of probiotics in aquaculture include the regulation of amino and fatty acid metabolisms, the excretion of digestive enzymes and vitamins or cofactors, the production of antagonistic compounds that inhibit bacteria, the enhancement of immune responses, the disruption of the quorum-sensing processes of pathogenic organisms, stress improvement, and heavy-metal detoxification.

#### 2.2.1. Probiotics Act as Growth Enhancers in Aquaculture

Probiotics play a crucial role in digesting complex dietary macronutrients. Additionally, they contribute to the host’s nutrient and vitamin supply and provide essential digestive enzymes, thereby enhancing feed utilization and digestion.

One of the mechanisms that regulates the metabolism of amino and fatty acids is the capacity of various probiotic strains to produce vitamin B12, as revealed by a study on carp guts [70,71]. In addition, this is helpful for enhancing fish growth and eradicating vitamin B12 deficiency in fish [72]. Also, essential macronutrients are usually supplied through feed. Various micronutrients such as amino acids, vitamins, and fatty acids are very important for physiological functions as nutrients in aquatic animals [73,74,75]. For instance, diverse fish species such as carp (*Cyprinus carpio*), rainbow trout (*Oncorhynchus mykiss*), channel catfish (*Ictalurus punctatus*), and tilapia (*Oreochromis niloticus*) have been found to synthesize vitamin B12 [76,77,78]. The growth and survival rates of juvenile black tiger shrimp (*Penaeus monodon*) were enhanced when they were fed for 100 days with a combination of *Lactobacillus* spp., previously isolated from the GITs of chickens [79]. In fact, probiotics improve the digestive function of aquatic animals by producing or inducing the secretion of different kinds of extracellular enzymes such as proteases, amylases, and lipases.

The function of probiotics results in abridged feed cost, which accounts for 60–70% of the contribution cost of fish production [80,81]. Both the maximum growth performance and best feed conversion ratio were detected when *O. niloticus* was fed with the probiotic *Micrococcus luteus* [82,83]. *Bacillus subtilis* improved feed digestibility; enhanced weight gain and feed conversion; and significantly increased the survival rate of bullfrogs (*Lithobates catesbeianus*) fed different doses (2.5, 5.0, and 10.0 g/kg) [84,85]. *Bacillus* species aid in the digestion of aquatic animals by supplying exoenzymes (proteases, lipases, and amylases) that enhance digestive enzymes [86]. The addition of probiotics (a mixture of *Streptococcus faecium, Lactobacillus acidophilus*, and *Saccharomyces cerevisiae*) at a concentration of 0.1% to Nile tilapia fry diets was found to enhance animal growth and intestinal alkaline phosphatase activity [87].

#### 2.2.2. Biocontrol of Bacterial Diseases in Aquaculture

In the past few decades, numerous studies have stated that probiotics synthesize different types of inhibitory substances responsible for antagonistic activity against pathogens. Two probiotic strains of LAB (*Lactococcus lactis* MM1 and *Enterococcus faecium* MM4) isolated from the intestine of the orange-spotted grouper (*E. coioides*) can secrete several inhibitory substances such as hydrogen peroxide and bacteriocin-like substances. These can be utilized to induce antimicrobial activity against different pathogens such as *Staphylococcus aureus*, *V. harveyi*, and *V. metschnikovi*, which affect groupers (*E. coioides*) [88,89]. The probiotic *B. pumilus* H2 has strong inhibitory activity against *Vibrio* spp. through its main mechanism of amicoumacin production, disrupting the cell membrane and cell lysis and thus showing anti-*Vibrio* activity [90,91]. The probiotic *Bacillus velezensis* cell-free supernatant contains different types of bioactive molecules that act against *A. salmonicida* infection [92]. The lipopeptide N3, synthesized by the probiotic *Bacillus amyloliquefaciens* M1, has strong antibacterial activity in the whole-cell membrane, which can exert significant effects from ion-conducting channels on the whole-cell membrane and membrane-active properties [93,94]. The probiotic species *Clostridium butyricum*, a culture supernatant, includes different types of inhibitory substances, mainly short-chain fatty acids (SCFAs); it can lower the pH of the intestine and thus decrease the growth of pathogens in fish intestinal epithelial cells [95]. The probiotic *E. faecium* was supplemented in the diets of Olive flounders and can enhance the antibacterial activity [96].

#### 2.2.3. Biocontrol of Viral Diseases in Aquaculture

Microorganism strains with potential probiotic effects in aquaculture such as *Pseudomonas* spp., *Vibrios* spp., and *Aeromonas* spp. induce antiviral effects against hematopoietic necrosis virus (IHNV) infection [97,98]. Similarly, the potential probiotic strain *Pseudoalteromonas undina* VKM-124 has been used to improve Yellow Jack (*Carangoides bartholomaei*) larval survival and enhance antiviral effects against Neuro Necrosis Virus (SJNNV) infections [99,100].

#### 2.2.4. Immunostimulant Agents in Aquaculture

Immunity development and modulation are among the various health benefits of probiotics in aquaculture. The majority of earlier studies have dealt with the health-boosting capabilities of probiotics in aquatic organisms. Currently, probiotics are significantly focused on the immunological development properties of the piscine immune system, including both innate and adaptive immunities [20]. Different types of probiotics improve various immunological properties, and notably, several fish use the efficiency of probiotics to vitalize teleost immunity in both in situ and ex situ conditions [101]. Although promising findings have been reported in previous studies, most immunostimulants do not progress to large-scale functions for fish. Since various immunostimulants in aquaculture produce similar effects, researchers have demonstrated the utilization of probiotics to enhance disease resistance and the immune system of carp fish species [102,103]. Several carp fish have shown an increase in the production of total serum protein, nitric oxide, lysozyme, albumin, and phagocytic activity via blood leucocytes; express IL-1b, superoxide anion, myeloperoxidase content, respiratory burst activity, and globulin levels; and complement C3, TNF-α, and lysozyme-C [102,104]. Current study reports indicate that probiotics (either single or mixed types) could enhance the immunological development of fish [105]. These reports have emphasized the immunomodulating properties of beneficial living cell organisms and the factors that facilitate the optimal induction of defense responses in the fish community. The probiotic strain *B. pumilus* SE5 has been isolated from the intestine of the fast-growing grouper, *E. coioides* [106,107], and subsequent studies have demonstrated that both viable and heat-inactivated *B. pumilus* SE5 could shape intestinal immunity and microbiota [108] and improve the growth performance and systemic immunity of *E. coioides* [109]. The dietary supplementation of the cell wall (CW), peptidoglycan (PG), and lipoteichoic acid (LTA) of the probiotic *B. pumilus* SE5 and its effect on intestinal immune-related gene expression and microbiota were evaluated in a 60-day feeding trial. The PG and LTA of the probiotic *B. pumilus* SE5 were more effective than the CW in shaping the intestinal immunity and microbiota of *E. coioides* [21], even though the mechanisms were largely unclear and needed further study.

#### 2.2.5. Interference of Quorum Sensing in Aquaculture

Quorum sensing (QS) is a communication system among bacterial cells that is very useful in controlling different kinds of biological macromolecule expressions like virulence agents in cell-thickness-dependent comparative performance [110]. In this process, QS bacteria produce and generate tiny marker molecules called auto-inducers [111]. The disruption of the QS process in pathogenic organisms is a potential anti-infective strategy, and different types of methods have been used to investigate QS. These include the inhibition of signal molecule biosynthesis, the application of QS antagonists, the chemical inactivation of QS signals with oxidized halogen antimicrobials, signal molecule biodegradation with bacterial lactonases and bacterial and eukaryotic acylases, and the application of QS agonists in aquaculture [112,113]. N-acyl homoserine lactones (AHLs) are the most important family of QS auto-inducers utilized in Gram-negative bacteria, and their biodegradation is a potential way to interrupt QS [114]. *Bacillus* species were among the first bacteria documented to degrade AHLs through the production of lactonase enzymes. Probiotic *Bacillus* strains can effectively secrete quorum-quenching enzymes and could reduce the pathogenic activity of *A. hydrophila* YJ-1 and control gut microbiota [115,116]. The dietary supplementation of probiotics with quorum-quenching activity has been shown to increase the intestinal barrier function and enhance the immune system of crucian carp against *A. hydrophila* infection. The quorum-quenching bacteria increase the expression of the tight junction (TJ) proteins ZO-1 and Occludin, which control the permeability and absorption of the intestinal mucosal barrier of crucian carp [117]. *Bacillus* sp. QSI-1 has been reported to be a quorum quencher in virulence agent production and the biofilm arrangement of the zebrafish pathogen *A. hydrophila*. In experimental trials, fish fed with *Bacillus* sp. QSI-1 exhibited a relative survival percentage of 80.8% [118]. In another study, AHL-degrading *Bacillus* sp. was shown to protect shrimp (*Penaeus monodon*) against *Vibrio harveyi* infection [119]. Furthermore, *Enterobacter* sp. f003 and *Staphylococcus* sp. sw120, isolated from fish intestines and pond sediment, respectively, have demonstrated the ability to degrade acyl-homoserine lactones (AHLs) and protect against *A. hydrophila* infection in the cyprinid *Carassius auratus gibelio* [120]. In a biofilm system, bacteria are resistant to high temperatures, phagocytic cells, surfactants, antibiotics, and antibodies and can alter their vital transmissions via quorum-sensing signaling [121]. These findings suggest that bacteria capable of degrading AHLs should be considered an alternative to antibiotics in aquaculture for effectively controlling bacterial infections in fish.

#### 2.2.6. Stress Improvement in the Aquaculture System

Stress in a fish’s life cycle disrupts all production. The cultured species may be weakened and averse to taking feed [122]. In this condition, probiotics in culture farms can decrease stress levels and help to enhance the innate immune system against pathogens and environmental stressors [123,124]. Probiotic treatments are very helpful in increasing the production of fish within the given time, and they also reduce the stress level in normal aquaculture practices. 

Studies have concluded that the use of some probiotic strains increases chronic stress resistance in zebrafish (*D. rerio*) [125,126]. Supplementation with an experimental nutritional probiotic, *Lactobacillus delbrueckii* sp. *Delbrueckii*, in sea bass led to a decrease in cortisol levels from 25 to 59 days, which, in fish tissue, is a stress indicator since it is directly engaged with the host’s reaction to stress [127]. One more approach evaluated how fish treated with probiotics exhibited increased flexibility in stress tests when compared with a control group [81]. The antioxidative properties of the probiotic *Lactobacillus fermentum* induce protective action in the intestinal microbial ecosystem and help to overcome exo- and endogenous oxidative stress [128]. The probiotic strain *Bacillus coagulans* SCC-19 alleviates the nonspecific immune damage induced by cadmium in common carp while also relieving oxidative stress induced by cadmium in fish [129].

#### 2.2.7. Reducing Heavy Metals in Aquaculture

Heavy metals such as lead (Pb), cadmium (Cd), silver (Ag), chromium (Cr), mercury (Hg), cobalt (Co), zinc (Zn), iron (Fe), and copper (Cu) are present in the soil, water, and atmosphere [130,131,132]. These metals can have toxic effects on all organisms and pose a huge risk to food quality, crops, and environmental quality. Heavy metals are mainly connected to anthropogenic action in the ecosystem [133]. Aqueous release from metal industries (steel, mining, and electroplating) contains elevated levels of heavy metals that end up in water bodies, and they are then also utilized for aquacultural action [134,135]. These heavy metals accumulate in fish tissue, and this is a matter of great concern with regard to humans consuming them via the food chain and breathing [133,135,136]. Their elimination is very helpful in reducing the toxic effects of the aquatic environment and outflow is, subsequently, imperative [137]. Among all the recommended methods of eliminating heavy metals is the process of utilizing microbes, which is cost-effective [138]. The action mechanisms of probiotics in detoxifying heavy metals can be classified into metabolically independent processes that do not require cellular energy, such as biosorption, and cellular-energy-dependent processes, namely, bioaccumulation and bioprecipitation [139].

Biosorption relies on a physicochemical process wherein cell-surface structures bind heavy metals through physical interactions. For example, *Lactobacillus acidophilus* and *Bifidobacterium angulatum* are effective in removing Cd, Pb, and As through electrostatic interactions between heavy-metal cations and the anionic functional groups of cell wall membranes [140]. Some probiotics release exopolysaccharides (EPSs), which can sequester heavy metals and reduce their bioavailability. The mechanisms underlying EPS-metal binding are mainly related to negatively charged acidic groups and steric structures on the surface of EPSs [141].

In bioaccumulation processes, probiotics accumulate heavy metals within their cells through energy-dependent processes. This can involve the synthesis and use of metal-binding proteins, such as metallothionein. For instance, *Bacillus cereus* can produce metallothionein in order to accumulate Pb [142].

Bioprecipitation involves the conversion of free metals into insoluble complexes, thereby reducing their bioavailability. Bacteria can catalyze oxidative and reductive processes to facilitate the precipitation of heavy metals. *Micrococcus* spp. have been demonstrated to be able to sequestrate heavy metals such as Zn, Cd, Pb, and Fe via calcite precipitation [143]

Generally, heavy metals activate the sporulation development of *Bacillus* species and thus decrease heavy metal absorption [134,144]. In addition, probiotic strains from aquatic farming sediments can be utilized as dietary supplements and help to remove heavy metals and metal-resistant microbes from the intestines of aquatic organisms, particularly fish, to control the progress of heavy metal accumulation [145].

### 2.3. Major Probiotic Genera as Biocontrol Agents in Aquaculture

The major probiotic genera used in aquaculture are *Lactobacillus* and *Bacillus* [146]. In most cases, *Bacillus*, *Lactobacillus*, *Lactococcus*, *Leuconostoc*, *Pediococcus*, and *Weissella* are isolated from fish and shellfish guts [147,148,149,150,151]. Supplementation in aquaculture feed is achieved using single-strain probiotics or associations of various bacteria as multi-strain probiotics (MSPs), which have been reported to have more beneficial effects on hosts owing to synergistic effects between various strains [152]. Table 2 lists some examples of probiotic-based functional feed additives for aquatic animals.

## 3. Prebiotics

Prebiotics are “non-digestible sugars, which helpfully influence the host by specifically enhancing the development of health-encouraging strains in the gut” [182,183]. Prebiotics improve the synbiotic association of the gut microbiota of the host [184] and are also known as immunosaccharides. There are various types of prebiotic compounds, including mannan oligosaccharide (MOS), fructooligosaccharide (FOS), and arabinooligosaccharide (AOS), all of which play a significant role in improving the natural immune system [185]. MOSs are most frequently used in animal diets. These prebiotics improve growth activity, feed utilization, survival rates, the development of immune reactions, and antagonistic activity against aquatic pathogens [186,187,188]. Oligosaccharide-type components have been connected to the development of immunity [189,190] and have been used extensively in diverse fish species such as *Psetta maxima* [13], *Larimichthys crocea* [191], *Paralichthys olivaceus* [192], *Rutilus rutilus* [193], *Piaractus mesopotamicus* [194], and *Acipenser Persicus* [195]. Previous study reports have examined the function of prebiotics in cultured finfish and shellfish, explaining that these compounds have significant effects on gut microbial composition, immune system, and infection resistance against pathogenic organisms in fish [196,197]. Previous studies have also verified the health-beneficial effects of prebiotics on growth and physiological status [198]. Prebiotics can improve the capability and feasibility of aquaculture production. The most frequently used prebiotics, including xylooligosaccharide (XOS), FOS, transgalactooligosaccharide (TGOS), glucooligosaccharide (GOS), soybean oligosaccharide (SBOS), polydextrose, inulin, and Lactosucrose, enhance aquaculture production [199]. Natural sources of prebiotics in vertebrates include onions, garlic, tomatoes, honey chicory, leeks, and so on [200].

### 3.1. Action in the Gastrointestinal Tracts of Aquatic Animals

Prebiotics exert possible effects on host biological responses, protecting fish species against harmful microbes and thus decreasing their mortality. However, an evaluation of the intestinal microbiota of important commercial fish like hybrid striped bass, channel catfish, salmonids, and tilapia is necessary to infer if there are any particular bacterial species that can be enhanced by the utilization of prebiotics. By increasing the production of volatile fatty acids (VFAs) in the GIT, the host’s advantage is the inhibition of potentially pathogenic organisms [201,202]. The synthesis of VFAs in the aquatic organism’s GIT indicates the presence of microbial communities [203]. Herbivorous fish were the first species (*Kyphosus cornelii* and *K. sydneyanus*) shown to contain VFAs synthesized by an intestinal bacterial community [204]. Another fish species, tilapia (*Oreochromis mossambicus*), was found to have VFAs produced by intestinal bacterial communities [205]. Prebiotics have numerous favorable effects on aquatic animals by enhancing disease resistance and improving nutrient accessibility [206]. Recently, our group evaluated the effects of FOS on the growth performance and predominant autochthonous intestinal microbiota of shrimp (*L. vannamei*) fed diets with fish meal partially replaced by soybean meal. The results showed that a dietary supplement of 2–4 g/kg of FOS could improve the growth performance and survival rate and exert a beneficial effect on the intestinal microbiota of shrimp. A dose adding 2–4 g/kg of FOS to shrimp diets with fish meal partially replaced by soybean meal was recommended [207,208].

### 3.2. Regulation in the Immune System of Aquatic Animals

In the past decades, prebiotics were used to regulate intestinal microbiota, modulate immunity, control pathogens, and increase the survival ability of aquatic animals, particularly fish such as sharks, rays, and bony fish [195]. Similar to all vertebrates, fish fully rely on their natural immunity against pathogens because of the restrictions on their adaptive immune functions [209]. There are various cellular and soluble components primarily concerned with immune responses, including phagocytes, leukocytes, and auxiliary cells, which are organized into tissues and organs, with leukocytes being the most functional. The impacts of prebiotics on immunity are indirect and involve the modification of gut microbes, thereby enhancing the immune system. Thus, these beneficial components assist in changing effectiveness, enhancing fish growth, and inducing inhibitory activity against pathogens by prohibiting linkage sites; natural organic acid (e.g., formic acid, lactic acid, acetic acid) syntheses; hydrogen peroxide; and numerous other compounds like bacteriocins, siderophores, lysozyme, and antibiotics. Through these action mechanisms, prebiotics can also cause changes in physiological and immunological responses in fish spleens, kidneys, and thymuses, which are major lymphoid organs [49,210]. The prebiotic components can act as growth promoters for commensal microbes by inhibiting the adhesion and assault of harmful microorganisms in epithelial cells. A beneficial effect of monosaccharide components arises, for instance, from enhancing immune functions, and it acts as a protection system for lymphoid organs.

#### 3.2.1. Phagocytosis

Phagocytosis is the process by which immune cells like macrophages and neutrophils engulf and digest foreign cells or particles, such as bacteria, viruses, and cellular debris [211]. FOS (0.5%) is used to enhance the phagocytosis, respiratory burst, and phenoloxidase activity of sea cucumber coelomocytes and infection resistance against *V. splendidus* infection [212]. The phagocytic capability of inhabitant and obtained trout macrophages are related to the circumstances (i.e., in suspension versus attached and spread) of the cells at the time of particle treatment. Substrate binding and cell spreading may play a very important function in controlling the overall phagocytic capabilities of macrophages. Since the host’s resistance against infectious agents depends upon the phagocytic ability of the cells, the finding that obtained trout macrophages can surround a larger number of activity latex particles than inhabitant cells provides a better understanding of immune regulatory mechanisms in fish [213]. Dietary supplementation with FOS significantly improves lysozyme activity compared with control diet groups. However, the phagocytic percentage of the phagocytic index has no significant effects. In addition, a combination of FOS and MOS (5.0 g/kg) has shown a significant difference in the phagocytic activity of Japanese flounders [195].

#### 3.2.2. Macrophage Activation

Macrophages play a very important role in the nonspecific and specific connections of immune function by synthesizing the highest level of immune reaction and eliminating harmful microbes. Macrophages are stimulated to produce diverse inflammatory cytokines like tumor necrosis factor (TNF), IL-1, IL-12, etc. [96]. The alterations to the physiology of macrophages as a result of environmental signals can benefit them with improved antimicrobial activity. Nevertheless, ecosystem signals do not always cause changes that improve macrophage immune activity. Both nonspecific and specific immune responses can result in macrophages that are more vulnerable to harmful infections and less prepared to generate cytokines that enhance immune system responses [214].

#### 3.2.3. Respiratory Burst Activity

A respiratory burst is the fast release of reactive oxygen substances, namely, superoxide anions, hydrogen peroxide, and hydroxyl radicals. These reactive oxygen compounds are generally used to defend the ability of the host organism to counter harmful microbes. They are synthesized by activated phagocytes that are responsible for destroying microbes [215]. Respiratory burst analyses have been performed in naturally resistant cells and blood neutrophils using the NBT (nitro blue tetrazolium) and MPO (myeloperoxidase) methods. Inulin (5 g kg^−1^) has been utilized as a dietary nutrient supplement for Nile tilapia and has improved lysozyme and hematocrit NBT action. It can also significantly enhance the natural immune system and increase the survival rate against *A. hydrophila* infection [216,217].

Marine invertebrates contain enzymes such as tyrosinases, laccases, and catecholases, which can be modified to complement the system of prophenoloxidase. This enhancement improves antagonistic activity through processes like phagocytosis and respiratory burst via opsonization. In a study conducted on red swamp crayfish, the supplementation of a prebiotic nutrient diet with 8 and 10 g kg^−1^ of FOS over a 30-day trial period significantly enhanced phenoloxidase reactions, stimulated immune-related genes (lysozyme, crustin 1, SOD), and increased the survival rate and antibacterial activity against *A. hydrophila* infection [218].

#### 3.2.4. Synthesis of Antibodies

B lymphocytes can produce special antibodies for recognizing specific microbial antigens, and these antibodies can neutralize antigens by surface binding and attaching to target cells. Prebiotics can stimulate the immune system like the production of antibodies. β-glucans, in particular, are known for their immunomodulatory effects because of their ability to bind to specific receptors on immune cells, such as macrophages, neutrophils, and natural killer cells, and enhance the release of signaling molecules such as cytokines. Such signaling molecules stimulate blood cells and enhance the secretion of antibodies that can recognize and bind to specific antigens (e.g., pathogens) [219,220]. The stimulation of antibody secretion (IgM) in crucian carp using glucans and astragalus polysaccharides as vaccine adjuvants has been demonstrated and has enhanced disease resistance against *A. veronii* [221]. The dietary supplementation of MOS and β-glucans was used to enhance the immune system of carp fry [222,223]. 

### 3.3. Major Prebiotics with Biocontrol Capabilities in Aquaculture

#### 3.3.1. β-Glucan

There is much evidence available regarding the positive effects of prebiotics on immune responses, disease resistance, and growth performance upon oral delivery in a variety of farmed animals such as salmonids [224], sea bream [225], and shellfish [226]. The supplementation of β-glucan as a prebiotic enhances growth activity and higher resistance action against pathogens in *P. vannamei* [227]. The prebiotic administration of β-glucans in diets is used to increase disease resistance; its efficiency depends on its origin and structure [228]. The glucan substance extracted from the cell walls of yeast (*S. cerevisiae*) has the ability to enhance the nonspecific immune system and disease resistance in Atlantic salmon [229].

#### 3.3.2. Oligosaccharide

Oligosaccharide components are crucial for the modulation of immune responses in fish species. The positive results of monosaccharide products have encouraged the development of various immunomodulating, environmentally friendly nutrient diet supplements for fish species [230]. Dietary supplementation with 1 to 1.5 g kg^−1^ of MOS is capable of improving the growth activity and the efficiency of common carp fingerlings, as well as their antibacterial resistance against *A. hydrophila* infections [231]. Nutrient feed additives (FOS) in beluga (*Huso huso*) juveniles have numerous beneficial effects such as gut microbiota modulation, immune response, digestive enzyme action, and growth performance [232]. Dietary supplementation with FOS at different concentrations (0%, 0.5%, and 1%) over 7 weeks in common carp has been proven to have significant effects on intestinal microbiota modulation and physiological response [233]. The dietary supplementation of MOS at 0.4% improves the growth performance and nonspecific immune responses of Asian catfish (*Clarias batrachus*) juveniles [234]. The prebiotic FOS, when used as a feed additive in juvenile large yellow croakers, has been found to improve growth action and digestive enzyme action [13,235]. 

Not all prebiotic substances have immunostimulant properties; only a few references are available regarding the effects of isomalto-oligosaccharide (IMO), which consists of a combination of isomaltotriose, isomaltose, panose, and isomaltotetraose, on aquatic animals. No clear statement has been recorded regarding immune responses [236].

#### 3.3.3. Chitosan

Chitosan is a linear polysaccharide component of β-(1–4)-linked D-glucosamine and is synthesized through alkaline deacetylation. It is a major component of arthropod exoskeletons, like those of shrimps, crabs, insects, and lobsters. In aquaculture, chitosan induces immunostimulation effects in various species, namely, rainbow trout [237], olive flounder (*Paralichthys olivaceus*) [238], and salmonids [239]. The administration of chitosan in the nutrient feed of *C. carpio* koi for 75 days resulted in significant effects such as an enhanced immune response, improved lipid metabolism, enhanced growth performance, and modulated intestine microbiota, thereby protecting the fish from pathogen invasion [240].

#### 3.3.4. Inulin

The prebiotic component inulin, a soluble plant fiber, is used in fish diets and plays a crucial role in enhancing the immune system in both mammals and fish. In aquaculture, inulin finds significant use by activating beneficial bacteria, inhibiting pathogens, and boosting immune system activity [241]. Inulin has the potential to mitigate inflammation induced by a high-carbohydrate diet, thereby enhancing pathogen resistance in fish. Additionally, supplementing with inulin leads to changes in gut microbiota composition and its metabolites. These alterations likely contribute to alleviating the metabolic syndromes induced by a high-carbohydrate diet in fish [242].

Figure 2 summarizes the main components of prebiotics from natural sources and their main action modes in improving host health. The functional feed additives of prebiotics in aquatic animals are summarized in Table 3.

## 4. Postbiotics

### 4.1. Concept, Definition, and Major Components of Postbiotics

The use of live microorganisms as probiotics may have potential issues associated with gene resistance acquisition and translocation and depends on their viability [245]. Likewise, it has been recognized that non-viable microorganisms, as well as their components and metabolites, can have positive effects on health, leading to the appearance of the postbiotic concept [246]. Postbiotics are defined by consensus panels as preparations of inactivated microorganisms and/or their components (cell fragments, cell walls, metabolites) that have beneficial health effects on hosts [247]. This definition does not include purified metabolites in the absence of cells or cell components. One definition defines postbiotics as dead microbes and/or cell structures or metabolites that are produced via bacterial lysis or secreted during the fermentation process [248].

Postbiotics include inactivated probiotics called paraprobiotics; metabolites like short-chain fatty acids (SCFAs), vitamins, and phenolic acids; secreted proteins and peptides; functional proteins and enzymes; cell wall components like LTAs and peptidoglycan (PG)-derived muropeptides; secreted and extracellular polysaccharides (EPSs); cell lysates; cellular components (glycans, enzymes); the microbial fraction; and surface molecules such as pili [249,250]. 

Figure 3 outlines the main postbiotic components.

### 4.2. Action Modes and Applications of Postbiotics in Aquaculture

The action mechanisms of postbiotics are still unclear, but it is generally assumed that they are similar to those of live probiotics [251]. Three main mechanisms are involved in postbiotic action modes.

#### 4.2.1. Immunomodulation via Microbial Compounds

Postbiotics act on the immune system through two signaling pathways, namely, nuclear factor-kB (NF-kB) and mitogen-activated protein kinase (MAPK), which are involved in immune and inflammatory responses. Postbiotics stimulate the innate and adaptive immune systems via external Toll-like receptors (TLRs), which recognize associated pathogens and bind to specific patterns such as LTAs and PGs. They also interact with intracellular nucleotide-like receptors (NLRs) and nucleotide-binding and oligomerization domain (NOD)-like receptors, which can bind to molecules like lipopolysaccharide (LPS), PG, and flagellin, thereby activating innate immune signaling pathways [248,250]. The role of PG recognition proteins in innate immune responses against pathogens has been demonstrated in fish [252,253]. PG-derived muropeptides from bacterial cell walls have been shown to boost the immune systems of fish [254] and shrimp [255]. For instance, muropeptides isolated from *Bifidobacterium thermophilum* have been proven to enhance shrimp immunity by increasing phagocytic activity or activating immune genes [255,256].

Additionally, postbiotics can enhance epithelial barrier protection via cell surface molecules such as pili and secreted protein P40 [257]. For example, the role of *Lactobacillus pentosus* surface protein on immune responses has been demonstrated in shrimp (*L. vannamei*) infected with *Vibrio parahaemolyticus* [258].

#### 4.2.2. Antagonizing Pathogens via Antimicrobial Activities

Postbiotics exhibit antimicrobial activities against various pathogens because of the presence of metabolites like peptides and organic acids [259]. Bacteriocin JFP2 isolated from *B. amyloliquefaciens* exhibits antimicrobial activity against the fish pathogen *A. hydrophila* [260]. The dietary addition of postbiotics containing LAB (*Lactobacillus*) has been reported to protect rainbow trout (*O. mykiss*) against the bacterial fish pathogen *L. garvieae* after 30 days of feeding [261].

#### 4.2.3. Inhibition of Oxidation via Antioxidant Enzyme Systems and Metabolites

Various postbiotics obtained from LAB have been shown to exhibit antioxidant activity, mainly attributed to phenolic compounds [262]. *L. plantarum* postbiotics have been documented to enhance antioxidant activity in animals [263]. In aquaculture applications, the overall antioxidant status of shrimp fed with diets supplemented with *C. butyricum* postbiotics was improved regarding an increase in alkaline phosphatase, acid phosphatase, total nitric oxide synthase, lysozyme, peroxidase, superoxide dismutase activities, total antioxidant capacity, and phenoloxidase content in the serum [264].

In aquaculture, postbiotics have been used as growth promoters instead of antibiotics, for immune system stimulation, and as disease control [257,265,266]. Recently, the potential application of postbiotics in aquaculture water quality in order to modulate bacterioplankton communities and influence nutrient cycling and bacterial pathogen abundance was reported [267]. Figure 4 illustrates the potential applications of postbiotics in aquaculture. Table 4 shows some recent potential applications of postbiotics in aquaculture.

## 5. Synbiotics

Synbiotics refer to dietary additives that blend probiotics and prebiotics in a synergistic combination, thereby enhancing their beneficial effects. When either dietary additives or supplements are used, the resulting positive effects typically follow one of three patterns: ingredient effects, synergism, or potentiation. Supplementation outcomes occur when the combined effects of both additives used together approximate the sum of the effects of the individual supplements. In the case of synergism, the amalgamated result of the two products is significantly greater than the sum of the effects of each factor administered alone. The term potentiation is used differently; some pharmacologists interchange it with synergism to describe a result that is better than that of a supplement alone, while others use it to describe an outcome that is only present when both substances are used simultaneously [273,274].

### 5.1. Possible Modes of Action of Synbiotics in Aquaculture

#### 5.1.1. Synbiotics Enhance Digestive Enzyme and Growth Performance

Dietary administration with synbiotics is helpful in enhancing the digestive enzymatic activities of fish, allowing the host to degrade more nutrients. This dietary method increases digestive action and likely enhances the weight gain rate and/or feed efficiency [275]. Nutrient diet supplementation with a mixture of probiotics and monosaccharides enhances feed efficiency and overall health in carp. However, limited data are available in aquaculture regarding the function of the nutrient diet supplementation of synbiotics in carp [24]. Nutrient diet administration with synbiotics enhances the lymphocytes and white blood cells in carp [276]. Synbiotics (IMBO), a combination of probiotics (*E. faecium*) and prebiotics (FOS), have been used to enhance the growth performance, survival rate, and digestive enzyme function of common carp fingerlings [277]. Dietary supplementation with FOS, MOS, and *B. clausii* can improve the growth performance and health benefits of the Japanese flounder more than a control diet [192]. Dietary supplementation with FOS and 1.35 × 10^7^ CFU g^−1^ *B. subtilis* (single or mixed) increases the specific growth rate (SGR) and feed efficiency ratio (FER) compared with the groups without *B. subtilis* additives in juvenile large yellow croakers (*Larimichthys crocea*) [235]. Figure 5 illustrates the possible modes of action of synbiotics in aquaculture.

#### 5.1.2. Synbiotics Improve Immune Response and Disease Resistance

An amalgamation of probiotic and prebiotic feed supplements is mainly helpful in enhancing the survival of beneficial organisms, as the presence of prebiotics protects well-organized fermentation. Finally, this rewards the host with a suitable approach [278]. The nutritional additives of probiotics and prebiotics (MOS, FOS, and inulin) enhance fish immune systems via the GIT [24,279,280]. A synbiotic composed of *Pediococcus acidilactici* and galactooligosaccharides improved immune parameters and antagonistic activity against *S. iniae* when administered to rainbow trout fingerlings for 8 weeks [281]. The combination of probiotic *Bacillus* spp. and 0.2% prebiotic isomaltooligosaccharide was used to improve immune functions in shrimp (*Penaeus japonicas*) against *V. alginolyticus* infection [282]. In addition, the blended use of *Bacillus* and molasses improved the microbial population and enhanced the development of the probiotic community and inhibitory activity against pathogens in Pacific white shrimp [283]. The effectiveness of a synbiotic treatment in conditions of defense against infectious factors can be evaluated with a confrontation examination given its regulatory power over harmful microbes and its capability to resist infections [284]. The functional feed additives of synbiotics in aquatic animals are summarized in Table 5.

## 6. Limitations of the Use of Synbiotic Agents in Aquaculture

The use of synbiotic agents in aquaculture instead of antibiotics has recently gained significant interest [290]. Probiotics have been shown to be effective in promoting growth, increasing immunity, and improving resistance to infections in aquatic animals [291]. The major limitation of their use comes from the problem of possible gene resistance acquisition and translocation, as well as the question of their viability and/or ability to colonize the fish gut [245]. The use of multi-strain probiotics increases the possibility of strain survival rates and, therefore, improves the beneficial effects on the growth, immunity, and infection resistance of aquatic animals [152]. Postbiotics present an advantage over probiotics because they do not have viability problems and are less susceptible to environmental conditions [245,292]. Additionally, they generally have a complex composition made up of several compounds that play multiple roles and have numerous beneficial effects on aquatic animals. However, their use in managing infectious diseases is still in its early stages [259].

Prebiotics, as inert biotic agents, are relatively safe and cost-effective alternatives to probiotics. Several studies on their immunostimulant properties and growth promotion in fish and shellfish have shown some evidence for their use in aquaculture [293]. Nevertheless, studies on the optimal dose should be carried out, as inadequate doses may lead to detrimental effects on aquatic animals [206,232]. Synbiotics improve the colonization of microorganisms in the intestines and are generally more effective than probiotics or prebiotics alone [292]. For example, Nile tilapia (*O. niloticus*) fed with synbiotics showed the highest increase in specific growth rate compared with a group fed with probiotics or prebiotics alone [276,294]. Extensive studies are still needed to specify the role of prebiotics, probiotics, postbiotics, and synbiotics in growth performance, intestinal health, and immune aspects with a focus on the mechanisms underlying the synbiotic diet in aquatic animals against various pathogens. The mode of administration and dose of the biotic agents are also important and certainly have an impact on their effectiveness [295].

The economic aspect of utilizing synbiotics and their components could be a limitation in aquaculture production. In the context of intensive aquaculture practices, the aspect of feeding comprises a substantial 60–80% of operational costs [296]. A Probiotic application in larval whiteleg shrimp (*L. vannamei*) resulted in a 6% increase in total production costs. However, the result of a higher survival rate contributed to a 44% reduction in unit production costs [297]. Studies on the feasibility of synbiotics in aquaculture have consistently shown improvement in economic efficiency compared with control diets, especially when aquatic animals have been under stress conditions such as high stocking density [298] or during the reproductive period [299].

## 7. Concluding Remarks and Future Perspectives

In conclusion, the aquaculture sector has experienced substantial growth in recent decades, confronting challenges related to environmental degradation and disease outbreaks, primarily because of the widespread prophylactic use of antibiotics and drugs. Synbiotic agents and their components, namely, probiotics, prebiotics, and postbiotics, emerge as natural and sustainable solutions considering their beneficial effects on growth performance, immunity, and overall health. These outcomes can be achieved by directly acting on aquatic animals through feeding or indirectly by improving the environment and water quality.

The direct-action mechanisms of these biotic family agents involve the modulation of the gut microbiota, leading to enhanced growth performance and feed utilization, as well as the reinforcement of the immune response, which helps aquatic animals resist pathogenic organisms. Indirectly, these natural solutions can assist in detoxifying the aquaculture system by removing heavy metals through biosorption, bioaccumulation, and bioprecipitation mechanisms, either through cellular-energy-dependent processes or not.

Moreover, these functional feed ingredients appear to be good alternatives to antibiotics and synthetic drugs given their multiple mechanisms of action in aquaculture, which help mitigate issues related to antibiotic resistance and the accumulation of harmful residues. While several study reports are available on probiotics, prebiotics, and synbiotics for the purpose of driving the development of aquaculture health and production, extensive studies are still needed at different levels for a deeper understanding of the mechanisms corresponding to the role of each component and combination in the growth performance, intestinal health, and immune aspects of aquatic animals. Furthermore, postbiotics, which are components or metabolites from dead probiotic microorganisms, such as functional amino acids, fatty acids, enzymes, exopolysaccharides, and organic acids, show promise as feed components because of their abilities to enhance the innate immune system, disease resistance, and growth and survival rates of aquatic animals.

Beyond the consideration of such biotic family agents and their combination with other functional ingredients such as herbs, it is also important to pay attention to combining biological solutions with other emerging technologies, such as nanoparticle-based delivery methods, in the future to improve efficiency in disease management, feeding formulation, and water quality.

## Figures and Tables

**Figure 1 biology-12-01498-f001:**
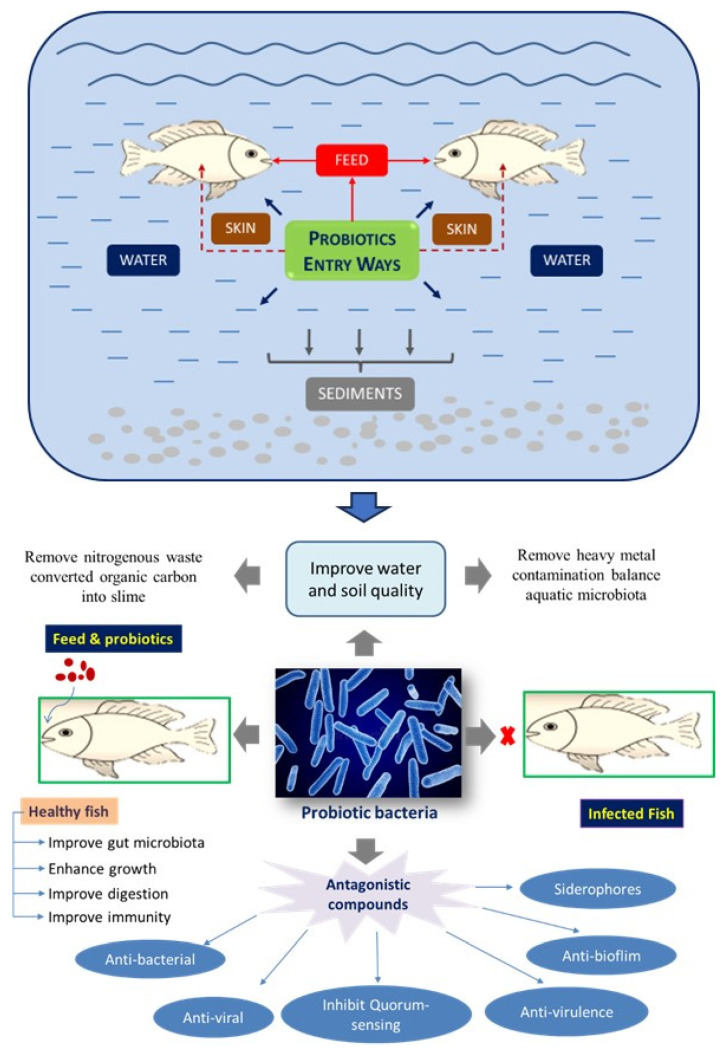
Illustration of the use and impact of probiotics in aquaculture systems.

**Figure 2 biology-12-01498-f002:**
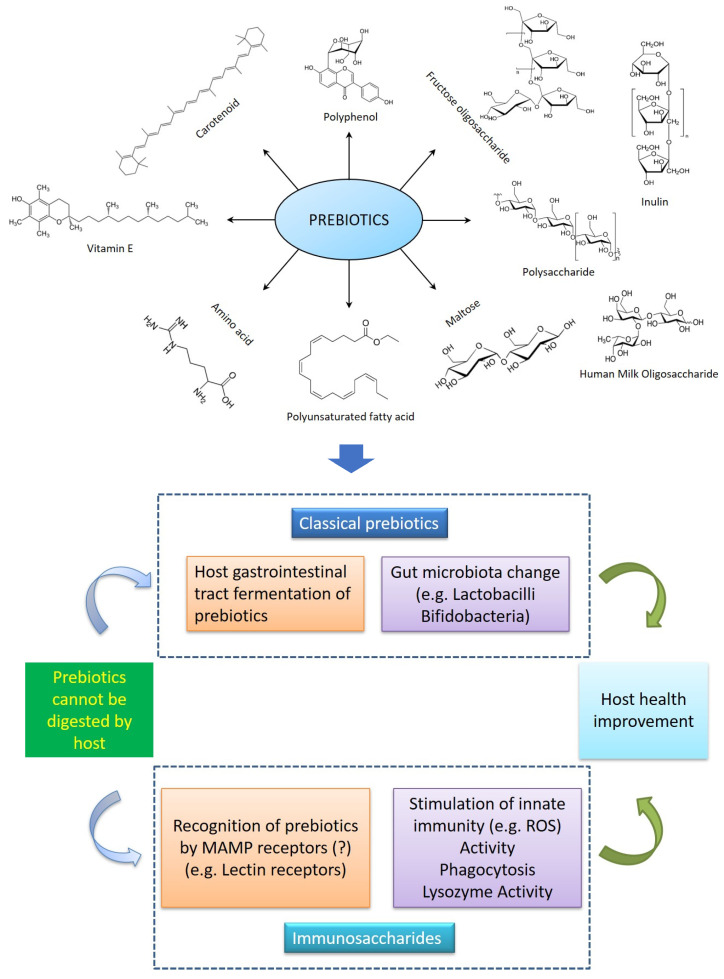
Main chemical components of prebiotics from natural sources and their action modes in improving host health.

**Figure 3 biology-12-01498-f003:**
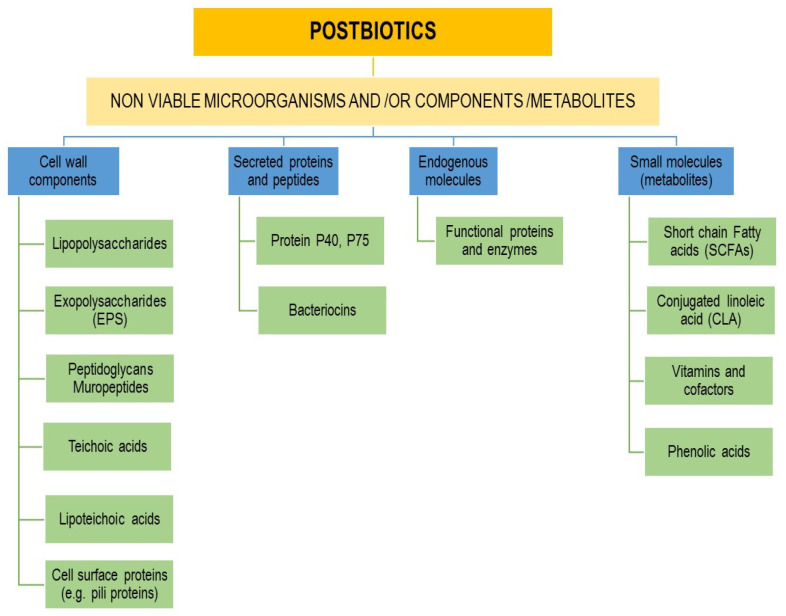
Postbiotic main components and molecules.

**Figure 4 biology-12-01498-f004:**
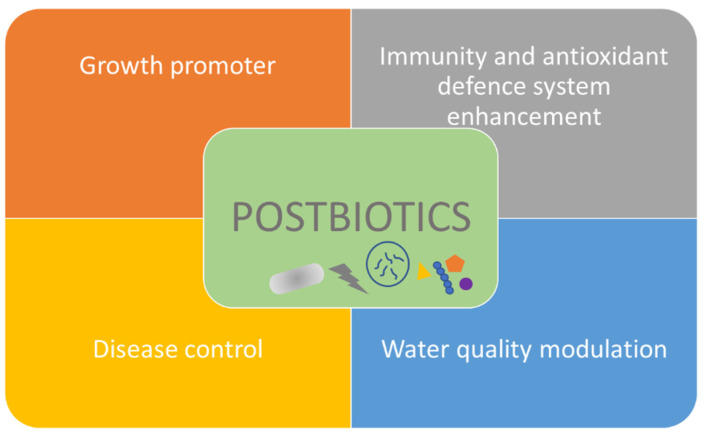
Postbiotics in aquaculture.

**Figure 5 biology-12-01498-f005:**
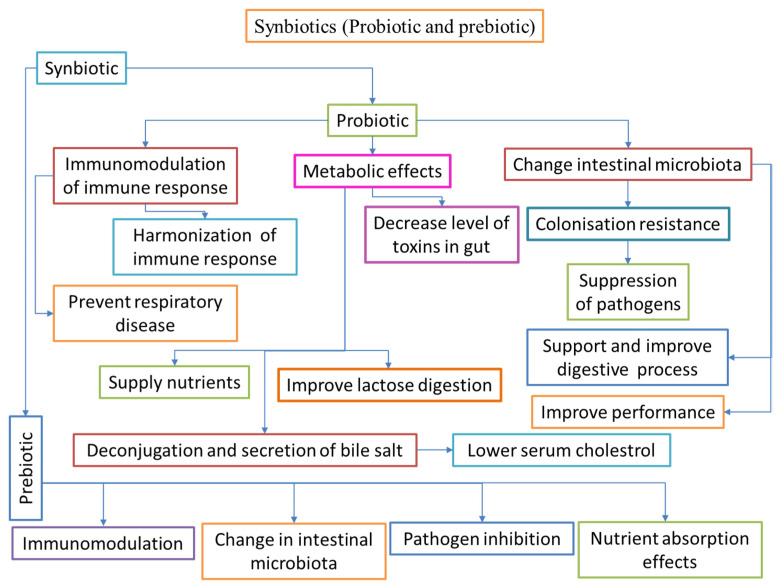
Illustration of modes of action of synbiotics in aquaculture.

**Table 1 biology-12-01498-t001:** A list of current probiotic strains for use in aquaculture.

Genus	Probiotics	Example of Target Fish Species	References
*Bacillus*	*Bacillus coagulans*	Common carp (*Cyprinus carpio*), turbot (*Scophthalmus Maximus*)	[52,53]
	*Bacillus subtilis*	Nile tilapia (*O. niloticus*)	[54]
	*Bacillus licheniformis*	Grass carp (*Ctenopharyngodon idella*)	[55]
	*Bacillus cereus*	Catfish (*Heteropneustes fossilis)*	[56]
*Bifidobacterium*	*Bifidobacterim bifidus*	Koi fish (*Cyprinus rubrofuscus*)	[57]
*Carnobacterium*	*Carnobacterium divergens*	Atlantic cod (*Gadus morhua*)	[58]
*Enterococcus*	*Enterococcus faecium*	Nile tilapia (*O. niloticus*)	[59]
*Lactobacillus*	*Lactobacillus casei*	Common carp (*Cyprinus carpio*)	[60]
	*L. plantarum*	Black sea bream (*Acanthopagrus schlegelii*)	[61]
	*L. rhamnosus*	Nile tilapia *(O. niloticus)*	[62]
*Lactococcus*	*L. lactis*	Mandarin fish *(Siniperca chuatsi)*	[63]
*Pediococcus*	*Pediococcus acidilactici*	Rainbow trout *(Oncorhynchus mykiss)*	[64]
*Streptomyces*	*Streptomyces sp.*	Zebrafish *(Danio rerio)*	[65]
*Saccharomyces*	*Saccharomyces cerevisiae*	Striped catfish (*Pangasianodon hypophthalmus*)	[66]
*Weissella*	*Weissella cibaria*	Common carp (*Cyprinus carpio*)	[67]

**Table 2 biology-12-01498-t002:** Functional feed additives of major probiotics in aquatic animals.

Probiotics Organisms	Functions	Aquatic Organisms	References
*Bacillus*			
*B. licheniformis* HGA8B	↑ growth performance and ↓ feed conversion ratio Up-regulation of immune genes	*O. niloticus*	[153]
*B. cereus* G19 *B. cereus* BC-01	↑ growth and immunity	*Apostichopus japonicus*	[154]
*B. cereus* EN25	Immunity and resistance against *Vibrio splendidus*	*A. japonicus*	[155]
*B. pumilus* SE5	↑ growth and immunity	*L. vannamei*	[156]
*B. subtilis* AB1	Bactericidal activity against *Aeromonas* infection	*O. mykiss*	[157]
*Bifidobacterium*			
*Bifidobacterium animalis* PTCC-1631	↑ growth performance, digestion, and nutrient utilization	*O. mykiss*	[158]
*B. lactis* PTCC-1736	↑ growth, nutrient digestibility, and carcass composition	*O. mykiss*	[158]
*Carnobacterium*			
*C. divergens* *C. maltaromaticum*	Antagonistic effects against *V. anguillarum*, *V. viscosus*, and *A. salmonicida*	*-*	[159,160]
*Lactobacillus*			
*L. plantarum* CLFP	↓ mortality against harmful strain *L. garvieae*	*O. mykiss*	[161]
*L. acidophilus*	Survival against *Staphylococcus xylosus, Aeromonas hydrophila* gr.2, and *Streptococcus agalactiae* infection	*Clarias gariepinus*	[162]
*L. pentosus*	↑ growth performance and feed conversion ratio ↑ survival against *Vibrio* species	*L. vannamei*	[163]
*Lactococcus*			
*Lactococcus lactis* BFE920	Activation of nonspecific immune system Bactericidal activity against *S. iniae*	*Paralichthys olivaceus*	[164]
*Leuconostoc*			
*Lc. Mesenteroides* CLFP 196	↑ survival against *A. salmonicida* infection	*Salmo trutta*	[165]
*Pediococcus*			
*P. pentosaceus* HN10	↑ feed utilization, digestive enzyme activity, and anti-Vibrio activity	*L. vannamei*	[166]
*Enterococcus*			
*E. casseliflavus* CGMCC1.2136	↑ growth performance, immunity, and digestive enzyme activity	*Rutilus rutilus caspicus*	[167]
*E. casseliflavus*	↑ growth performance and disease resistance against *S. iniae*	*O. mykiss*	[168]
*E. durans*	↑ growth performance and survival rate	*O. mykiss*	[169]
*Clostridium*			
*C. butyricum*	↑ antibacterial activity against *Vibriosis* infection	*O. mykiss*	[170]
*C. butyricum*	↑ immunity; regulation of gut microbiota; antagonistic effects against *Aeromonas* sp., *Vibrio* sp., and *Pseudomonas* sp.	*C. carpio*	[171]
*Weissella*			
*W. confusa*	↑ growth performance	*O. mykiss*	[172]
*W. confusa*	↑ growth performance and antibacterial activity against *A. hydrophila*	*Lates calcarifer*	[173]
Other strains			
*A. veronii* BA-1	↑ immune system and antibacterial activity	*C. carpio*	[174]
*Micrococcus luteus*	↑ growth performance and feed conversion ratio	*O. niloticus*	[175]
*Pseudoalteromonas undina* VKM-124	↑ survival and antiviral activity	*Carangoides bartholomaei*	[99]
Yeast			
*S.cerevisiae*	↑ growth performance and resistance against waterborne Cu toxicity	*Sarotherodon galilaeus*	[176]
*S. cerevisiae*	↑ immunity and ↓ mortality against *P. fluorescens*	*Mystus cavasius*	[177]
*Yarrowia lipolytica*	↑ immune response, antioxidant status, and disease resistance against *V. parahaemolyticus* infection	*Lutjanus peru*	[178]
Multi-strain			
*B. subtilis* and *Bacillus licheniformis* (BioPlus2B)	↑ resistance against *Y. ruckeri*	*O. mykiss*	[179]
*Lactobacillus delbrueckii* *Lactobacillus rhamnosus* *L. plantarum* *B. bifidum*	↑ growth performance and immunity	*Acipenser baerii*	[180]
*Lactobacillus plantarum* (STBL1), *Saccharomyces cerevisiae* (STBS1), and *Bacillus safensis* (SQVG18)	↑ growth, antioxidant capacity, digestion, and gut microflora	*P. vannamei*	[181]

*↓* decrease or reduction; ↑, increase or improvement.

**Table 3 biology-12-01498-t003:** Functional feed additives of prebiotics in aquatic animals.

Prebiotics	Functions	Aquatic Species	References
FOS	↑ growth, survival, and gut microbiota section	*L. vannamei*	[208]
β-glucan	↑ growth, survival, and immune system	*Sparus aurata*	[225]
MOS	↑ growth, immune system, antioxidant capacity, and intestinal health	*Cyprinus carpio*	[243]
Chitosan	↑ growth, feed utilization, lipid metabolism, gut microbiota composition, and immune system	*Cyprinus carpio koi*	[240]
Inulin	↑ growth, antioxidant capacity, immunity, and gut microbiota at low salinity	*L. vannamei*	[244]

↑, increase or improvement.

**Table 4 biology-12-01498-t004:** Some recent potential applications of postbiotics in aquaculture.

Postbiotics	Microorganism Producer	Aquatic Species	Applications	References
Exopolysaccharides	*Lactococcus lactis* Z-2	Common carp (*C. carpio*)	Immunity enhancement Resistance against *A. hydrophila*	[268]
Cell surface proteins	*L. pentosus*	Shrimp (*Litopenaeus vannamei*)	Immune response improvement	[258]
Cell wall components (PGs and LTA)	*B. pumilus* SE5	Grouper (*E. coioides*)	Growth performance improvement Innate and adaptive immunity amelioration	[109]
Lipoteichoic acids	*L. plantarum* LTA	Silvery pomfret (*Pampus argenteus*)	Resistance against *V. anguillarum*-caused vibriosis	[269]
Non-living microorganisms	*S. cerevisiae*, *B. velezensis* and *Cetobacterium somerae*	Common carp (*C. carpio*)	Gut microbiota improvement Enhancement of nonspecific immunity Antioxidant status improvement	[270]
	Dried autolyzed yeast	Gilthead sea bream (*Sparus aurata*)	Intestinal microbiota improvement	[271]
	*Rhodotorula minuta* and *Cetobacterium somerae*	Hybrid sturgeon (*Acipenser baerii* × *Acipenser schrencki*)	Growth performance improvement Nonspecific immunity improvement	[265]
	Heat-killed *L. plantarum* L-137	Nile tilapia (*O. niloticus*)	Growth performance stress resistance and immunity enhancement	[272]

**Table 5 biology-12-01498-t005:** Functional feed additives of synbiotics in aquatic animals.

Synbiotics	Functions	Aquatic Organisms	References
*P.acidilactici* + GOS	↑ growth, survival, and digestive enzyme function	*Labidochromis lividus*	[285]
*B. clausii* + FOS, MOS	↑ growth, survival, and digestive enzyme function	*Paralichthys olivaceus*	[286]
*P.acidilactici* + GOS	↑ immunity and antagonistic activity against *S. iniae* infections	*Oncorhynchus mykiss*	[287]
*B. subtilis* + *L. acidophilus* + *S. cerevisiae* + FOS	↑ growth and feed efficiency ratio	*Eriocheir sinensis*	[288]
*P.acidilactici* + IMO	↑ growth, immune response, and antioxidant capacity	*C. carpio*	[289]

↑, increase or improvement.

## Data Availability

Data sharing not applicable.

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
