# Peer review of "Synbiotic Agents and Their Active Components for Sustainable Aquaculture: Concepts, Action Mechanisms, and Applications"

_biology, 2023, doi:10.3390/biology12121498_

Round 1

Reviewer 1 Report

Comments and Suggestions for Authors

The present review is in good shape, quite extensive and very interesting, it covers the functional feed additives possess several beneficial char- acteristics, including gut microbiota modulation, immune response reinforcement, resistance to pathogenic organisms, improved growth performance, and enhanced feed utilization in aquatic animals.  

Line 44. Species or genus should be mentioned

Line 50. Reference is needed for his statement

Line 52. Correct spaces

Line 53. Aren´t crustaceans or any other invertebrate aquatic animal more exposed to conditions, the Word “especially” generates noise

Line 53 – 62. Will be ideal to mention the phtagonens in relation with the species, according to the animals you are talking about earlier in the ms; tilapia, catfish, pangasius

Line 68-75. In order to stablish your idea, facts od specific cases of studies are needed. Mention the studies about impacts of chemoterapeutics

Line 160-163 the paragraph is too short, more explanation about the subject is needed

Line 178-179 how is that probiotics provided growth nutrients (pathways)

Line 186-188 provided more facts and information about the research

Line 220 define piscine system

Line 276 reference is needed for the concept food irrational fear

Line 276 I prefer feed instead of food 

Line 299 301 you should mentioned studies about heavy metals accumulation in aquaculture

Line 304 312 the statement about probiotics and heavy metals accumulation is not quite explained, more discussion is needed

Line 487 as prebiotics were described, same should be with probiotics

Line 638 a deeper explanation of the economics limitations in the use of biotics should me mentioned

More concrete and integrated conclusions are suggested 

Author Response

Dear Reviewer,

We would like to thank you for having considered and spent your time to read and comment our manuscript ID: biology-2720981. 

On behalf of all Authors, I am pleased to submit the revised version, which takes into account all of your comments and suggestions. All changes and corrections made in the revised manuscript have been highlighted.

You will also find below the responses to your comments/questions.

Sincerely,

Hary Razafindralambo (PhD)

Reviewer 1

  • From Line 44 to Line 62. “Species or genus should be mentioned; Line 50. Reference is needed for his statement; Line 52. Correct spaces; Line 53. Aren´t crustaceans or any other invertebrate aquatic animal more exposed to conditions, the Word “especially” generates noise; Line 53 – 62. Will be ideal to mention the phtagonens in relation with the species, according to the animals you are talking about earlier in the ms; tilapia, catfish, pangasius”

According to these remarks and those of the second reviewer, the two first paragraphs of the manuscript have been simplified, as follows:

“Aquaculture is an emerging  sector that generates numerous employment opportunities and also addresses a fundamental need for essential nutrients in global food production [1]. However, it is presently faced to pressing challenges, especially the vulnerability of aquatic animals to ecological degradation and infectious outbreaks. A key contributing factor to these threats is the excessive use of antibiotics and synthetic drugs, which exert harmful effects on the aquatic environment [2,3].

  • Line 68-75. In order to stablish your idea, facts on specific cases of studies are needed. Mention

the studies about impacts of chemotherapeutics

This part has been revised with providing references and examples, as follows:

“However, some of these chemotherapeutic applications have been widely criticized due to their negative impacts on marine debris gathering, drug resistance expansion, and immunosuppressant activity. For example, the use of formalin and potassium permanganate for pathogen control has resulted in adverse effects on fish like damage in gills (hyperplasia) and alteration in mucous cells [5,6]. The extensive application of antibiotics in aquaculture have led to their bioaccumulation in aquatic animals.”

  • Line 160-163 the paragraph is too short, more explanation about the subject is needed

This paragraph has been completed to introduce the different subsections described in the section 2.2. Possible modes of action of probiotics in aquaculture, as follow:

Significant effects of probiotic, e.g. Bacillus sp. as feed supplements, include the improvement of growth performance, digestive enzyme activity, resistance to pathogens, and immune response in aquatic animals [68,69]. Possible action modes of probiotics in aquaculture include the regulation of amino and fatty acids metabolisms, excretion of digestive enzymes and vitamins or cofactors, production of antagonistic compounds that inhibit bacteria, enhancement of immune responses, disruption of the quorum sensing process of pathogenic organisms, stress improvement, and heavy metal detoxification.

  • Line 178-179 how is that probiotics provided growth nutrients (pathways)

This phrase has been removed to avoid redundance (Line 167-171)

  • Line 186-188 provided more facts and information about the research

Details have been provided with citing appropriate reference, as follows:

“The addition of probiotics (Mixture of Streptococcus faecium and Lactobacillus acidophilus, and Saccharomyces cerevisiae) at a concentration of 0.1% to Nile tilapia fry diets enhances animal growth and intestinal alkaline phosphatase activity [87].”

  • Line 220 define piscine system

A short definition has been added:

“Currently, probiotics are significantly focused on the immunological development properties of the piscine immune system, including both innate and adaptive immunities”.

  • Line 276 reference is needed for the concept food irrational fear

The phrase “The concept food irrational fear” has been removed because judged useless.

  • Line 276 I prefer feed instead of food

The correction has been done.

  • Line 299 301 & Line 304 312: you should mentioned studies about heavy metals accumulation in aquaculture and the statement about probiotics and heavy metals accumulation is not quite explained, more discussion is needed.

Further explanations and discussion about the heavy metals accumulation in aquaculture have been added to this section, as follows:

“The action mechanisms of probiotics in detoxifying heavy metals can be classified into metabolically independent processes that do not require cellular energy, such as biosorption, and cellular energy-dependent processes namely bioaccumulation and bioprecipitation [139].

Biosorption relies on a physicochemical process wherein cell surface structures bind heavy metals through physical interactions. For example, Lactobacillus acidophilus and Bifidobacterium angulatum are effective in removing Cd, Pb and As through electrostatic interactions between heavy metals cations and anionic functional groups of cell wall membranes [140]. Some probiotics release exopolysaccharides (EPS), which can sequester heavy metals and reduce their bioavailability. The mechanisms underlying EPS-metal binding are mainly related to the negatively charged acidic groups and steric structure on the surface of EPS [141].

In Bioaccumulation processes, probiotics accumulate heavy metals within their cells through energy-dependent processes. This can involve the synthesis and use of metal-binding proteins, such as metallothioneins. For instance, Bacillus cereus can produce metallothioneins to accumulate Pb [142].

Bioprecipitation involves the conversion of free metals into insoluble complexes, thereby reducing their bioavailability. Bacteria can catalyse oxidative and reductive processes to facilitate the precipitation of heavy metals. Micrococcus sp. has been demonstrated to able to sequestrate heavy metals such as Zn, Cd, Pb, and Fe via calcite precipitation [143].”

  • Line 487 as prebiotics were described, same should be with probiotics

Additional information on probiotics including new and recent references has been listed in Table 1.

  • Line 638 a deeper explanation of the economics limitations in the use of biotics should me mentioned

A paragraph has been added by explaining the economic limitations of using biotic family agents, as follows:

“The economic aspect of utilizing synbiotics and their components could  be a limitation in aquaculture production. In the context of intensive aquaculture practices, the aspect of feeding comprises a substantial 60-80% of operational costs [298]. The probiotic application for larval whiteleg shrimp (L. vannamei) resulted in a 6% increase of total production costs. However, the result of a higher survival rate contributed to a 44% reduction in unit production costs [299]. Studies on the feasibility use of synbiotics in aquaculture have consistently shown improvement on the economic efficiency compared to a control diet, especially when aquatic animals were under stress conditions such as high stocking density [300], or during the reproductive period [301].”

  • More concrete and integrated conclusions are suggested.

The conclusion has been improved with more concrete and integrated statements, as follows:

“In conclusion, the aquaculture sector has experienced substantial growth in recent decades, while confronting challenges related to environmental degradation and disease outbreaks, primarily due to the widespread prophylactic use of antibiotics and drugs. The utilization of synbiotic agents and their components, namely probiotics, prebiotics, and postbiotics, emerges as natural and sustainable solutions considering their beneficial effects on growth performance, immunity, and overall health. These outcomes are achieved by directly acting on aquatic animals through feeding or indirectly by improving the environment and water quality.

The direct action mechanisms of these biotic family agents involve the modulation of the gut microbiota, leading to enhanced growth performance and feed utilization, as well as reinforcement of the immune response that helps aquatic animals to resist to pathogenic organisms. Indirectly, these natural solutions can assist in detoxifying the aquaculture system by removing heavy metals through biosorption, bioaccumulation, and bioprecipitation mechanisms, either through cellular energy-dependent processes or not.

Moreover, these functional feed ingredients appear as good alternatives to antibiotics and synthetic drugs due to their multiple mechanisms of action in aquaculture, which help mitigate issues related to antibiotic resistance and the accumulation of harmful residues. While several study reports are available on probiotics, prebiotics, and synbiotics to drive the development of aquaculture health and production, extensive studies are still needed at different levels for a deeper understanding of the mechanisms corresponding to the role of each component and their combination in growth performance, intestinal health, and immune aspects of aquatic animals. Furthermore, postbiotics, which are components or metabolites from dead probiotic microorganisms, such as functional amino acids, fatty acids, enzymes, exopolysaccharides, and organic acids, show promise as feed components due to their capacities in enhancing the innate immune system, disease resistance, and growth and survival rates of aquatic animals.

Beyond the consideration of such biotic family agents and their combination with other functional ingredients such as herbs, it is also important to pay attention to combining biological solutions with other emerging technologies, such as nanoparticle-based delivery methods in the future, to improve efficiency in disease management, feeding formulation, and water quality.”

Reviewer 2 Report

Comments and Suggestions for Authors

This paper aims to review different aspects associated with the application of probiotics, prebiotics and synbiotics in aquaculture. There are several previously published review papers in this area, and the authors need to specify the criteria considered to include the papers listed in this paper and update their paper with more focus on recent mechanisms of action described for probiotics and prebiotics. This paper fails to cover and discuss novel and recent findings related to the topics provided in the review with enough details (topics are too general). The following major corrections should also be made or addressed in the revised version:

The title of this paper should be changed as it is misleading and only probiotics, among other mentioned topics, are biotic agents.

There are several repetitious and general sentences in this review, which need to be corrected and more details are suggested to be included.

The first two paragraphs (lines 42-62) should be removed or replaced by relevant paragraphs.

Line 66. "nitrofurans" should be listed in antibiotics (lines 64-65).

Line 100, 162. Please revise "infection resistance".

Line 101. Please replace "consisting in" with "consisting of".

Line 118. Please revise "adherence to probiotics".

Line 122,124. Please revise "might be managed " and "can be managed ".

Line 145. predators ??

Lines 155-158. Please rewrite.

Line 185, 283. "Bacillus" and "Delbrueckii" should be italic.

Lines 200, 205. Please specify which types of bioactive molecules or inhibitory substances.

Line 245. There are several papers published on the potential of AHL-degrading (quorum quenching) probiotics for protection of fish against bacterial pathogens, none of which has been cited in this paper.

Lines 304-306. Please rewrite.

Line 311. categories ??

Line 321. In table 1, ↓feed conversion ratio.

Lines 378-388. Please rewrite.

Line 434. This is too general and there is no information about induction of synthesis of antibodies in this paragraph.

Line 461. Please revise "to get better".

Line 465. They ??

Line 572. Please replace "probiotics" with "postbiotics".

Line 597. "digestive enzymatic activities" is correct.

Comments on the Quality of English Language

Please see the comments.

Author Response

Dear Reviewer,

We would like to thank you for having considered and spent your time to read and comment our manuscript ID: biology-2720981. 

 On behalf of all Authors, I am pleased to submit the revised version, which takes into account all of your comments and suggestions. All changes and corrections made in the revised manuscript have been highlighted.

 You will also find below the responses to your comments/questions.

 Sincerely,

 Hary Razafindralambo (PhD)

Reviewer 2

There are several previously published review papers in this area, and the authors need to specify the criteria considered to include the papers listed in this paper and update their paper with more focus on recent mechanisms of action described for probiotics and prebiotics. This paper fails to cover and discuss novel and recent findings related to the topics provided in the review with enough details (topics are too general).

Thank you so much for this remark.

The paper has been revised accordingly by reminding the basic mechanisms of action (e.g., modulation of the gut microbiota to enhance growth performance and feed utilization) and providing details on recent mechanisms, among others those related to the immunomodulation activities that are described for probiotics (section 2.2.4), prebiotics (section 3.2), postbiotics (section 4.2.1), and the combination of probiotics and prebiotics, i.e., synbiotics (section 5.1.2). References have been updated according to such a revision.  

Major corrections

  • The title of this paper should be changed as it is misleading and only probiotics, among other mentioned topics, are biotic agents.

The title of the paper has been changed as follows:

Synbiotic Agents and their Active Components for Sustainable Aquaculture: Concepts, Action Mechanisms and Applications

In the previous version, we referred as “biotic agents” the biotic family agents related to the terminology pro-, pre-, syn-, and post-biotic.

  • There are several repetitious and general sentences in this review, which need to be corrected and more details are suggested to be included.

The manuscript has been revised according to these remarks: detected repetitions have been removed (e.g., line 44-62, line 178-179, etc.) and more details have been included (e.g., 2.2.5. Interference of quorum sensing in aquaculture: line 257-273; 2.2.7. Reducing heavy metals in aquaculture: line 308-328).

  • The first two paragraphs (lines 42-62) should be removed or replaced by relevant paragraphs. 

The first two paragraphs have been simplified, as follows:

“Aquaculture is an emerging  sector that generates numerous employment opportunities and also addresses a fundamental need for essential nutrients in global food production [1]. However, it is presently faced to pressing challenges, especially the vulnerability of aquatic animals to ecological degradation and infectious outbreaks. A key contributing factor to these threats is the excessive use of antibiotics and synthetic drugs, which exert harmful effects on the aquatic environment [2,3]”.

  • Line 66. "nitrofurans" should be listed in antibiotics (lines 64-65).

Done

  • Line 100, 162. Please revise "infection resistance".

 “…Infection resistance…” has been replaced by “…resistance to pathogens,…”

  • Line 101. Please replace "consisting in" with "consisting of".

 Done

  • Line 118. Please revise "adherence to probiotics".

 “…Adherence to probiotics…” has been replaced by “…probiotic adhesion capacity…”

  • Line 122,124. Please revise "might be managed " and "can be managed ".

Done

  • Line 145. predators ??

This word has been replaced by pathogens.

  • Lines 155-158. Please rewrite.

The corresponding paragraph has been rewritten, as follows:

“Potential probiotic strains are assessed based on physiological, functional, and safety criteria such as stress resistances (e.g., acid and bile tolerance), gut epithelial adherence, survival rates, pathogen inhibiting activities, large-scale cultivability, no hemolytic activity, non-pathogenicity, , absence of plasmid-encoded antibiotic resistance genes,  and beneficial effects on host animals. These include, for instance, their capacity as growth promoters, and anti-inflammatory, antimutagenic, and immunostimulatory agents. Every new strain used for probiotic expansion mainly contain all the aforesaid features [28,50,51].”

  • Line 185, 283. "Bacillus" and "Delbrueckii" should be italic.

Done

  • Lines 200, 205. Please specify which types of bioactive molecules or inhibitory substances.

The main type of bioactive molecules or inhibitory substances such as short chain fatty acids (SCFA) has been specified.

  • Line 245. There are several papers published on the potential of AHL-degrading (quorum quenching) probiotics for protection of fish against bacterial pathogens, none of which has been cited in this paper.

The paragraph related to the AHL-degrading probiotics has been detailed and some references have been added, as follows:

“In experimental trials, fish fed with Bacillus sp. QSI-1 exhibited a relative percentage survival of 80.8%.  [118]. In another studies, AHL-degrading Bacillus sp. protects shrimp (Penaeus monodon) against Vibrio harveyi infection [119]. Furthermore, Enterobacter sp. f003 and Staphylococcus sp. sw120, isolated from fish intestines and pond sediment respectively, demonstrated the ability to degrade acyl-homoserine lactones (AHLs) and protect against A. hydrophila infection in the cyprinid Carassius auratus gibelio [120].”  

  • Lines 304-306. Please rewrite.

The corresponding paragraph has been rewritten, as follows:

“The action mechanisms of probiotics in detoxifying heavy metals can be classified into metabolically independent processes that do not require cellular energy, such as biosorption, and cellular energy-dependent processes namely bioaccumulation and bioprecipitation [139].

Biosorption relies on a physicochemical process wherein cell surface structures bind heavy metals through physical interactions. For example, Lactobacillus acidophilus and Bifidobacterium angulatum are effective in removing Cd, Pb and As through electrostatic interactions between heavy metals cations and anionic functional groups of cell-wall menbranes [140]. Some probiotics release exopolysaccharides (EPS), which can sequester heavy metals and reduce their bioavailability. The mechanisms underlying EPS-metal binding are mainly related to the negatively charged acidic groups and steric structure on the surface of EPS [141].

In Bioaccumulation processes, probiotics accumulate heavy metals within their cells through energy-dependent processes. This can involve the synthesis and use of metal-binding proteins, such as metallothioneins. For instance, Bacillus cereus can produce metallothioneins to accumulate Pb [142].

Bioprecipitation involves the conversion of free metals into insoluble complexes, thereby reducing their bioavailability. Bacteria can catalyze oxidative and reductive processes to facilitate the precipitation of heavy metals. Micrococcus sp. has been demonstrated to able to sequestrate heavy metals such as Zn, Cd, Pb, and Fe via calcite precipitation [143].”

  • Line 311. categories ??

“…antibiotic-resistant categoriess…” has been replaced by “…, metal-resistant microbes…”

  • Line 321. In table 1, ↓feed conversion ratio.

The arrow has been added.

  • Lines 378-388. Please rewrite.

This paragraph has been revised, as follows:

“Thus, these beneficial components assist in changing effectiveness, enhancing fish growth, and inducing inhibitory activity against pathogens by prohibiting linkage sites, natural organic acid (e.g., formic acid, lactic acid, acetic acid) syntheses, hydrogen peroxide, and numerous other compounds like bacteriocins, siderophores, lysozyme, and antibiotics. Through these action mechanisms, prebiotics can also cause a change in physiological and immunological responses in fish’s spleen, kidney and thymus, which are major lymphoid organs [49,212]. The prebiotic components can act as growth promoter for commensal microbes by inhibiting the adhesion and assault of harmful microorganisms in the epithelial cells. A beneficial effect of monosaccharide components arises, for instance, from enhancing immune function and acting as a protection system for lymphoid organs as well.”

  • Line 434. This is too general and there is no information about induction of synthesis of antibodies in this paragraph.

This paragraph has been revised, as follows:

“B lymphocytes could produce special antibodies for recognizing specific microbial antigens and these antibodies could neutralize the antigens by surface binding and attaching to the target cells.  Prebiotics can stimulate the immune system like the production of antibodies. b-glucans, in particular, are known for their immunomodulatory effects due to their ability to bind to specific receptors on immune cells such as macrophages, neutrophils and natural killer cells, and enhance the release of signaling molecules such as cytokines. Such signaling molecules stimulate blood cells and enhance the secretion of antibodies that can recognize and bind to specific antigens (e.g., pathogens) [221,222]. Stimulation of antibodies secretion (IgM) in crucian carp using glucans and astragalus polysaccharides as vaccine adjuvant has been demonstrated and enhanced disease resistance against A. veronii [223]. The dietary supplementation of MOS and β-glucans were used to enhance the immune system of carp fry [224,225].” 

  • Line 461. Please revise "to get better".

This paragraph has been revised as follows:

“The diet supplementation of MOS from 1 to 1.5g kg-1 was capable to improve the growth activity and the feed efficiency of common carp fingerlings, as well as their antibacterial ability against A. hydrophila infection [233]. The nutrient feed additives (FOS) in beluga (Huso huso) juveniles had numerous beneficial effects such as gut microbiota modulation, immune response, and digestive enzyme action and growth performance [234]. Dietary supplementation of FOS at  different concentrations (0%, 0.5%, and 1%) during 7 weeks in common carp has been proved to have significant effects on intestinal microbiota modulation and physiological response [235].”

  • Line 465. They ??

The phrase “They also presented a 7-week…” has been replaced by “Dietary supplementation of FOS at different concentrations (0%, 0.5%, and 1%) during 7 weeks in common carp has been proved to have significant effects on intestinal microbiota modulation and physiological response [235].”

  • Line 572. Please replace "probiotics" with "postbiotics".

Done

  • Line 597. "digestive enzymatic activities" is correct.

Done

Round 2

Reviewer 2 Report

Comments and Suggestions for Authors

The authors have addressed the revisions asked, and this review paper can now be published in the journal.